# Assessment of Platelet Aggregation and Thrombin Generation in Patients with Familial Chylomicronemia Syndrome Treated with Volanesorsen: A Cross-Sectional Study

**DOI:** 10.3390/biomedicines12092017

**Published:** 2024-09-04

**Authors:** Ilenia Lorenza Calcaterra, Renata Santoro, Nicoletta Vitelli, Ferdinando Cirillo, Guido D’Errico, Cornelia Guerrino, Giovanna Cardiero, Maria Donata Di Taranto, Giuliana Fortunato, Gabriella Iannuzzo, Matteo Nicola Dario Di Minno

**Affiliations:** 1Department of Clinical Medicine and Surgery, Federico II University of Naples, 80131 Naples, Italy; renata.santoro@studenti.unina.it (R.S.); nicoletta.vitelli@unina.it (N.V.); ferdinando.cirillo@unina.it (F.C.); guidoderrico1@gmail.com (G.D.); corneliaguerrino7@gmail.com (C.G.); gabriella.iannuzzo@unina.it (G.I.); matteo.diminno@unina.it (M.N.D.D.M.); 2Department of Molecular Medicine and Medical Biotechnology, Federico II University of Naples, 80131 Naples, Italy; giovanna.cardiero@unina.it (G.C.);

**Keywords:** familial chylomicronemia syndrome, severe hypertriglyceridemia, volanesorsen, antisense oligonucleotide, platelets, thrombin generation

## Abstract

Background: The antisense oligonucleotide against APOC3 mRNA volanesorsen was recently introduced to treat Familial Chylomicronemia Syndrome (FCS). Cases of decreased platelet count are reported among patients treated with volanesorsen. The aim of the study was to evaluate platelet function and thrombin generation (TG) assessment in FCS patients receiving volanesorsen. We performed a cross-sectional study on FCS patients treated with volanesorsen. Methods: Changes in platelet count PLC were assessed from baseline to Tw12 and Tw36. To assess TG, samples were processed by CAT (with PPP-reagent LOW). The results were expressed by the thrombogram graphic (thrombin variation over time); LagTime; endogenous thrombin potential (ETP); peak; time to reach peak (ttpeak), StartTail and Velocity Index. Platelet aggregation was assessed by testing different agonists using the turbidimetry method. Results: Four FCS patients and four matched healthy controls were included in the present study. Changes in PLC were 30% at Tw12 and 34% at Tw36. Thrombin generation results showed values in the normal range (for patients and controls, respectively, LagTime:10.42 ± 4.40 and 9.25 ± 0.99; ttPeak:14.33 ± 4.01 and 13.10 ± 0.67; StartTail: 32.13 ± 3.54 and 29.46 ± 1.69; Velocity Index: 20.21 ± 3.63 and 33.05 ± 13.21; ETP: 599.80 ± 73.47 and 900.2 ± 210.99; peak value: 76.84 ± 1.07 and 123.30 ± 39.45) and no significant difference between cases and controls. Platelet aggregation test showed values in range, with no significant difference compared to healthy controls. Conclusions: Our study showed for the first time that no significant changes in general hemostasis assessed by TG and in platelet function were observed in FCS patients receiving volanesorsen.

## 1. Introduction

Familial Chylomicronemia Syndrome (FCS) is a rare monogenic disease (prevalence of 1–9:1,000,000) with autosomal recessive transmission. FCS is caused by loss-of-function mutations in the lipoprotein-lipase (*LPL*) gene or in LPL co-factor genes such as apolipoprotein A5 (*APOA5*), lipase maturation factor 1 (*LMF1*), glycosylphosphatidylinositol-anchored high-density lipoprotein-binding protein 1 (*GPIHBP1*) and glycerol-3-phosphate dehydrogenase 1 (*G3PDH1*) [1].

Although available lipid-lowering therapies (fibrates, statins and omega-3 fatty acids) decrease serum triglycerides [2,3], a non-negligible percentage of subjects with hypertriglyceridemia (HTG), particularly those with FCS, do not achieve therapeutic goals [4].

Targeting Apo-CIII is a promising approach to treat refractory HTG and to prevent acute pancreatitis in FCS subjects [5]. Apo-CIII is a glycoprotein (consisting of 79 amino acids) that is synthesized mainly by liver and to a lesser proportion by bowel. Apo-CIII is involved in different steps of triglyceride (TG) metabolism. Apo-CIII is linked to triglyceride-rich lipoprotein (TRLs), including chylomicrons and very-low-density lipoprotein (VLDL) particles, as well as high-density lipoprotein (HDL) particles [6]. Apo-CIII may be involved in regulating TG exchange between HDL and TRLs with a mechanism that is still unknown. Additionally, Apo-CIII is an inhibitor LPL enzyme and is able to prevent TRL remnant uptake by hepatic lipoprotein receptors. The second-generation antisense oligonucleotide (ASO) against APOC3 mRNA volanesorsen (previously called ISIS 304801, ISIS-ApoCIIIRx and IONIS-ApoCIIIRx) was recently introduced to treat HTG [5,6,7,8].

Volanesorsen is a second-generation 2′-O-(2-methoxyethyl)–modified antisense inhibitor of Apo-CIII synthesis. Inhibition of Apo-CIII synthesis in the liver occurs through sequence-specific binding of volanesorsen to APOC3 mRNA, which in turn induces the degradation of APOC3 mRNA by the ribonuclease RNase H1 [6].

A recent metanalysis including four RCTs showed a ≈ 92% mean reduction in TG in subjects receiving volanesorsen as compared to placebo-treated controls, accompanied by a significant reduction of 70% in Apo-B48 levels [5]. Cases of decreased platelet count (PLC) were reported during treatment with volanesorsen. The underlying mechanism of this side effect is still unknown, and further studies are needed to explore a potential change in platelet function and the overall hemostatic process.

Literature data showed that volanesorsen is well tolerated, and the thrombocytopenia was mainly mildly dose-related and reversible. In view of this, volanesorsen received approval from the Food and Drug Administration (FDA) for FCS treatment [9].

More in detail, a phase 3, double-blind, randomized, 52-week study (APPROACH) conducted on a group of 66 individuals with FCS, with the primary outcome being the change in fasting TG percentage before and after 3 months of treatment, reported adverse effects mainly including injection site reactions and thrombocytopenia. A PLC lower than 100,000 per microliter (μL) was reported in 16 out of 33 treated subjects, including two with a count lower than 25,000 per μL, with no bleeding occurring in any of the cases [10]. For this reason, the patients were treated with immunoglobulins and steroids with close platelet monitoring, accompanied by a dose reduction, and achieved complete recovery [10].

However, it has been shown that thrombocytopenia is not just an ASO-related adverse event, as changes in platelet count are often reported in FCS patients [11]. Indeed, in a study conducted on homozygotes for variants in the LPL gene causing FCS (HoLPL), on heterozygotes for variants causing loss of function (HeLPL), and on a control group (WT), it was shown that in fasting condition, patients with FCS and HeLPL had a lower PLC than the normolipidemic control group [12]. Following meal consumption, PLC rose slightly in the control group, whereas in FCS patients, the levels remained low even 5 h after consumption, and this drop was more pronounced in HoLPL than in HeLPL. These changes in PLC have been shown to correlate with plasma TG levels, neutrophil/lymphocyte ratio (NLR) and lymphocyte count, but not with platelet activity [12]. Although FCS patients show a lower PLC than healthy subjects, platelet activity seems not to be affected [11].

To further evaluate the key aspect of hemostasis and to observe and to estimate the impact of a potential adverse effect, the aim of our study was to assess platelet function by platelet aggregation test and thrombin generation assessed by the Calibrated Automated Thrombogram (CAT) test in patients with FCS, currently treated with volanesorsen.

## 2. Materials and Methods

### 2.1. Study Population

Patients attending the Regional Reference Center for the Treatment of Dyslipidemia at the “Federico II” University Hospital were recruited from January 2022 to September 2023, applying the following inclusion and exclusion criteria.

Inclusion criteria: diagnosis of FCS; eligibility for treatment with volanesorsen, with reference to the criteria identified by the Italian Medicines Agency (AIFA), including genetic diagnosis of FCS, previous episode of acute pancreatitis in the last 5 years; TG > 885 mg/dL despite hypolipidic diet, fibrates and polyunsaturated fatty acids (PUFA); basal PLC > 140,000 platelets per μL; and stable treatment with volanesorsen (>12 weeks).

Exclusion criteria: age < 18 years; incompetence to understand or sign the consent form; high transaminase level (greater than 3 × upper limit of normal); decompensated diabetes mellitus (HbA1C > 8%); end-stage renal failure (glomerular filtration rate < 30 mL/min/m^2^); ongoing neoplasm or diagnosis of malignant tumor within 2 years prior to first visit; secondary HTG (induced by hypothyroidism, corticosteroids, or hormone therapy).

### 2.2. Study Protocol

The patients gave their written informed consent to participate in the study, and all data of interest were entered anonymously into a special database. Following informed consent, a detailed medical history was taken for each patient: data were collected on age, sex, previous and/or current medical conditions, and current and previous lipid-lowering therapy with dosage and frequency of administration. Body weight was assessed using an electronic beam scale with digital readout and to the nearest 0.1 kg, following bladder emptying. Patients were measured barefoot and in light indoor clothing.

The study was approved by our ethics committee (protocol 2015/261) in accordance with the Declaration of Helsinki, and the remaining patients and guardians of the participants provided informed consent.

Before the start of the study, all enrolled patients received lipid-lowering therapy (fibrate and high-dose PUFA) and implemented treatment with volanesorsen 285 mg (vials) administered subcutaneously every 7 days for the first 3 months (induction dosage) and then once every 14 days as maintenance.

### 2.3. Hematology Blood Parameters

Periodic blood count monitoring (including PLC) was performed according to AIFA indication for patient monitoring during treatment with volanesorsen.

In the frame of the present cross-sectional study, we evaluated PLC on the same day of evaluation with other coagulative tests. We also evaluated changes in PLC from baseline values (before starting volanesorsen treatment) to week 12 (after concluding the induction dose of volanesorsen) and to week 36 during treatment with volanesorsen.

### 2.4. Thrombin Generation

TG in plasma was measured with the calibrated automated thrombogram (CAT) assay as developed by Hemker and co-workers [13]. CAT was evaluated at week 36 during treatment with volanesorsen.

Thrombin generation was expressed based on endogenous thrombin potential (ETP); LagTime (LT); thrombin peak (TP), and time-to-thrombin peak (TTP).

### 2.5. Platelet Aggregation

Platelet aggregation was evaluated by the Born method of turbidimetric aggregation [14] using a CHRONO-LOG^®^ 700 Series Whole Blood/Optical Lumi-Aggregometer with AGGRO/LINK^®^ 8 Software at week 36 during treatment with volanesorsen. Platelet-rich plasma (PRP) was tested with four agonists (ADP, arachidonic acid, collagen and ristocetin) and compared with a sample of platelet-poor plasma (PPP). The test was executed in two channels, adding two different concentrations for each agonist (1.8 μM and 4.0 μM ADP, 0.4 µg/mL and 0.8 µg/mL collagen, 0.8 mg/mL and 1.25 mg/mL ristocetin). Only arachidonic acid was tested once, at 0.4 mM. Platelet aggregation induced by addiction of each agonist was evaluated through the percentage of light transmission: ≥50% within 3 min indicates no impairment of platelet aggregation. This test is effective in analyzing platelet function abnormalities.

### 2.6. Statistical Analysis

Statistical analysis of the data was performed using the IBM SPSS 29 System (SPSS Inc., Chicago, IL, USA). Continuous data were expressed as mean value ± standard deviation (SD) or, in the case of variables with non-normal distribution, as median IQR range.

To compare continuous variables for paired and independent samples, the *t*-test was performed; in the case of values with a non-Gaussian skewed distribution, the Mann–Whitney U test was used. Categorical variables were expressed in % and analyzed with the χ^2^ test or Fisher’s exact test in the case where the minimum expected value was <5. Correlations were assessed using Pearson’s linear correlation coefficient (r), and in the case of non-parametric variables, Spearman’s correlation (rho) was calculated. All results were expressed as two-tailed values, with *p*-value < 0.05 considered to be statistically significant.

## 3. Results

We enrolled four FCS patients (reported by initials of first and last name, CG, CL, BA, PS), two men and two women, all diagnosed with FCS and treated with volanesorsen, and four healthy volunteers as control group, matched for gender and age.

### 3.1. Study Population

The clinical and demographic characteristics of the study population are shown for individual patients in Table 1 and as cumulative comparisons with controls in Table 2. Overall, they had a mean age of 56.5 ± 0.5 years and a mean BMI of 22.1 ± 1.89. Three patients (BA, CG, CL) were carriers of the HoLPL variant. CG and CL, who were related, presented the same variant, secondary to inbreeding on the parents’ side. PS was a carrier of pathogenic variants in the APOA5 cofactor in compound heterozygosity.

All patients were eligible for treatment with volanesorsen according to the AIFA criteria. All patients were treated with a hypolipidic diet plus MCT oil (20 g/day). and lipid-lowering drugs. In detail: micronized fenofibrate 145 mg for CL, BA and PS; gemfibrozil 600 mg 2 cp per day for CG; and PUFA (6 g/day).

All patients had developed pancreatitis during their lifetimes; CG reported an episode during pregnancy. Patient BA presented more than 20 episodes of acute pancreatitis. Both CL and BA presented evolution towards chronic pancreatitis resulting in exocrine pancreas deficiency.

All patients complained of chronic abdominal pain and, except for BA, developed diabetes secondary to pancreatitis.

### 3.2. Effects of Volanesorsen on Platelet Count

PLC was evaluated according to the AIFA recommendations for patient monitoring every 2 weeks. PLC at baseline had a median value of 237.5 platelets/μL (IQR: 228.0–304.0); after 12 weeks, the median PLC value was 181.0 platelets/μL (IQR: 159.0–214.8), and at week 36, it was 178.5 platelets/μL (IQR: 162.0–211.8). All patients showed a PLC in the reference range (150,000–450,000 platelets/μL) at baseline. We observed a 30% reduction in PLC from baseline to week 12 and a 34% reduction from baseline to week 36. However, none of patients reported PLC values lower than 100,000 platelets/μL, and no clinically relevant bleeding episodes were reported. Individual and cumulative percent changes in PLC are reported in Figure 1a,b, respectively.

### 3.3. Thrombin Generation Analysis

The thrombin generation analyses shown in Table 3 resulted in mean LagTime values of 10.42 ± 4.40; ETP of 599.8 ± 73.47; peak of 76.84 ± 1.07; ttPeak of 14.33 ± 4.01; StartTail of 32.13 ± 3.54; and Velocity Index of 20.21 ± 3.63. All values were within the estimated normal ranges and did not differ from those of the healthy controls, whose values are shown in Table 3. However, comparing the cases and the controls, we could observe an interesting difference in the amount of thrombin obtained, i.e., the ETP (599.80 ± 73.47 for the cases and 900.2 ± 210.99 for the controls) and the maximum concentration of thrombin obtained, i.e., the peak (76.84 ± 1.07 for the cases and 123.30 ± 39.45 for the controls), as can be seen in Table 3. This could also be verified graphically in the thrombograms obtained (Figure 2). However, these differences were not statistically significant.

### 3.4. Platelet Aggregation Analysis

The platelet aggregation tests showed values in range: ADP (1.8 μM) 62%; Arachidonic Acid (0.4 mM) 66%; Collagen (0.8 μg/mL) 84%; Ristocetin (1.25 mg/mL) 90% for patients, as compared to heathy controls with ADP (1.8 μM) 70%; Arachidonic Acid (0.4 mM) 72%; Collagen (0.8 μg/mL) 85%; Ristocetin (1.25 mg/mL) 92%. No significant difference was found between cases and controls

## 4. Discussion

Results of our study show that volanesorsen treatment induced a mild reduction in PLC (never lower than 100,000 platelets/μL) in four FCS patients. These results are in line with those obtained from clinical trials that evaluated the efficacy and safety of volanesorsen [5]. Accordingly, in none of the patients was a dose-delayed administration needed. This effect on PLC reduction has also been documented for other second-generation ASOs [15]. However, the pathophysiological mechanism of this side effect is still unclear. One of the possible hypotheses is that the class of ASOs may promote the binding of neutrophils to platelets [12]. To the best of our knowledge, there are no studies aimed at evaluating the impact of ASO-induced reduction of PLC on platelet aggregation and global hemostasis. Our study provides insight into hemostatic function beyond the clinical observation of PLC and hemorrhagic manifestations. The results of thrombin generation assessment revealed values within the estimated normal range and showed no significant differences from healthy controls.

Platelet aggregation was performed in only one patient, due to the impossibility of performing the proposed method with lactescent samples. It showed normal results when compared to healthy controls, thus detecting an optimal status of global hemostasis.

To date, this represents the first study aimed at exploring platelet function and thrombin generation capacity in FCS patients on volanesorsen therapy. Previous data from the literature show that PLC fluctuations are quite common in patients with FCS secondary to pathogenic variants in the LPL gene.

A recent study showed that fluctuations in PLC assessed over 15 years were reported in FCS patients that are carriers of LPL variants. It was observed that about 55% reported PLC < 150,000 platelets/μL at least once. Paradoxically, it was also observed that approximately 12% presented thrombocytosis, with a PLC > 450,000 platelets/μL, and 3.6% presented both thrombocytosis and thrombocytopenia [11]. Furthermore, a more recent study showed that post-prandial fluctuations in PLC are present in FCS patients that are carriers of homozygous variants in the LPL gene not affecting platelet function [12]. In our study, patients carrying the pathogenic variant in the *LPL* gene were found to have lower PLC values than patients carrying the pathogenic variants in the *APOA5* gene.

We did not observe significant differences between cases and healthy controls in the thrombin generation results. However, globally analyzing the results even in a small sample of patient carriers of *LPL* gene variants, we found lower values in thrombin generation analysis but still within a normal range.

From a physiological point of view, the LPL enzyme is bound to endothelial cells via heparan sulphate proteoglycans (HSPGs), which makes it possible to identify the luminal surface of endothelial cells as the site of catalytic action [16]. Heparan sulphate (HS) is known to inhibit thrombin generation by inducing the inactivation of factor Xa and thrombin, via antithrombin III [17]. The interaction between LPL and HSPGs has the effect of inhibiting platelet adhesion and activation [18]. Thus, an *LPL* gene variant that compromises the physiological interaction with HSPGs could influence platelet function. Exploring the impact of genotype on LPL bound to HSPGs in our study population using the gnomAD and UniProt databases, in patients CL and CG, the c.844G > T homozygous variant (p.Glu282*) located in exon 6 of the LPL gene encodes for a stop codon, leading to the production of a truncated protein that lacks the interaction domain with HS present in the C-terminal portion of LPL [19]. For patient BA, the homozygous variant c.621C > G (p.Asp207Glu) is located in exon 5 of the LPL gene and encodes for a missense variant that we cannot establish with certainty has an effect on the interaction with HSPGs, as it does for CL and CG; it is only known that this variant impairs the lipolytic activity of the enzyme [19,20].

Patient PS is compound heterozygous for the c.427delC p.(Arg143Alafs*57) variant in exon 4 and c.49 + 5G > A in intron 2 of the APOA5 gene. APOA5 is able to facilitate the interaction between HSPGs and TRLs, thereby increasing the activity of LPL [21]. The variant in exon 4 results in the production of a truncated protein that would not have part of a 42 amino acid positive stretch (residues 186–227), postulated in one study as a site of interaction with HSPGs.

Considering this, it could be assumed that reduced binding of LPL to HSPGs may favor the exposure of HS chains, thereby inhibiting thrombin generation. Similarly, a pathogenic variant in the APOA5 gene could have an effect similar to that seen for LPL variants, albeit to a lesser extent. This may explain how, although generation is within the estimated normal ranges, a slight variation in the amount of thrombin obtained and a slight shift in the peak were observed. In fact, the HTG condition would not affect thrombin generation, as demonstrated in a study carried out on patients with hypertriglyceridemia [22]. From this point of view, our study shows an exploratory role in an area where there are grey areas in the literature; therefore, further studies are needed to further investigate this clinical question.

The main limitation of our study is represented by the small sample size. However, FCS is an extremely rare disease (1:1,000,000), and therefore, our sample is representative of disease prevalence in our region according to Italian epidemiological data. Another limitation derives from the potential influence of chylomicrons on the platelet aggregation test. However, results were entirely comparable to those obtained in healthy controls. It would be interesting to perform the same analyses in patients reporting more severe PLC reduction to explore the presence of functional changes in this setting. However, the extreme rarity of the disease and the low incidence of the side effect makes this difficult.

## 5. Conclusions

Our study shows for the first time that in patients treated with volanesorsen, despite a modest reduction in PLC, no significant alterations were observed in platelet function in terms of platelet aggregation and general hemostasis studied by means of the thrombin generation test. It is inferred that factors related to the underlying pathology, especially the genotype, may contribute to such manifestations that are not just a class effect of the drug. This study represents a starting point for a translational exploration of the global hemostatic pattern of patients with FCS. Therefore, further studies with larger study samples will be needed to better define this clinical issue.

## Figures and Tables

**Figure 1 biomedicines-12-02017-f001:**
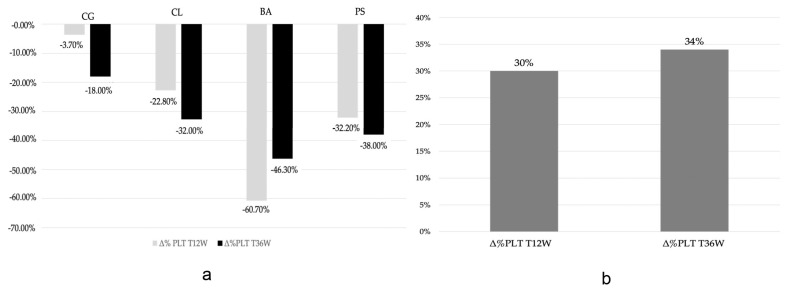
Individual percent changes in PLC from baseline to T12w and T36w (**a**). Cumulative percent changes in PLC from baseline to T12w and T36w (**b**).

**Figure 2 biomedicines-12-02017-f002:**
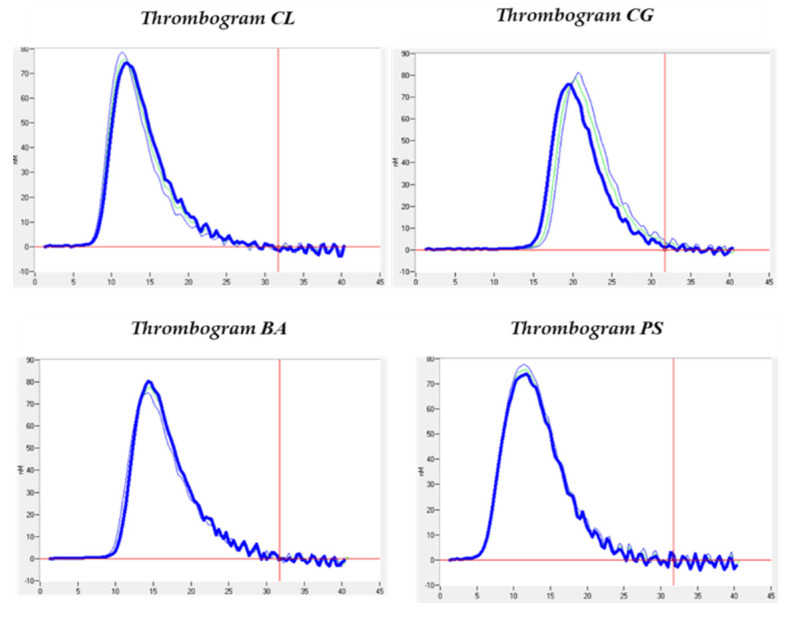
Outcome of thrombin generation (CAT) shown for each patient.

**Table 1 biomedicines-12-02017-t001:** Clinical and demographic characteristics of each patient (CG, CL, BA and PS).

Patient	CG	CL	BA	PS	Mean ± SD
Age (years)	62	71	53	40	56.5 ± 13.2
Sex *(M = 1)*	0	1	1	0	N.A.
BMI (kg/m^2^)	20.8	22	24.8	20.8	22.1 ± 1.9
Age at diagnosis	20	22	6	4	13 ± 9.3
Pathogenic variants	LPL c.844G > T (p.Glu282 *) stop gained	LPL c.844G > T (p.Glu282 *) stop gained	LPL c.621C > G (p.Asp207Glu) missense	APOA5 c.427delC p.(Arg143Alafs*57) c.49 + 5G > A	
Previous pancreatitis	4	10	20	3	9.25
Pancreatitis in last 5 years	1	2	10	2	3.75
Chronic pancreatitis (yes = 1)	0	1	1	0	50%
Diabetes	1	1	0	1	75%
Treatment (fibrates, PUFA and MCT oil)	Gemfibrozil 600 mg bid, PUFA (6 g), MCT oil	Fenofibrate 145 mg/daily, PUFA (6 g), MCT oil	Fenofibrate 145 mg/daily, PUFA (6 g) MCT oil	Fenofibrate 145 mg/daily, PUFA (6 g) MCT oil	–

*: translation termination (stop) codon.

**Table 2 biomedicines-12-02017-t002:** Clinical and demographic characteristics of patients and controls.

Parameter	FCS Patients (*n* = 4)	Controls (*n* = 4)	*p*-Value
Age (years)	56.5 ± 13.2	55.4 ± 11.3	0.899
Sex (% males)	50%	50%	–
Weight (kg)	57.5 ± 9.5	58.3 ± 8.2	0.758
BMI (kg/m^2^)	22.1 ± 1.9	23.1 ± 2.1	0.856
Waist circumference (cm)	78.0 ± 2.5	79.0 ± 1.9	0.924
Platelet count (PLC/μL)	178.5 (IQR: 162–211.8)	240.5 (IQR: 232–247.8)	0.098
Triglycerides (mg/dL)	874 (IQR: 701.3–1000.5)	93.5 (IQR: 83.5–101.5)	**<0.001**

**Table 3 biomedicines-12-02017-t003:** Thrombin generation analysis in FCs patients vs. controls.

Parameter	FCS Patients	Controls	*p*-Value
LagTime	10.42 ± 4.40	9.25 ± 0.99	0.685
ETP	599.80 ± 73.47	900.2 ± 210.99	0.102
Peak	76.84 ± 1.07	123.30 ± 39.45	0.104
ttPeak	14.33 ± 4.01	13.10 ± 0.67	0.620
StartTail	32.13 ± 3.54	29.46 ± 1.69	0.143
Velocity Index	20.21 ± 3.63	33.05 ± 13.21	0.149

Statistical analyses of thrombin generation (CAT) results in patients and healthy controls expressed as mean ± SD. Parameters analyzed are LagTime (indicates time interval between addition of trigger and thrombin generation), ETP (endogen thrombin potential indicates how much thrombin has been generated), peak (the maximum thrombin concentration obtained), ttPeak (time to peak indicates thrombin generation’s velocity), StartTail (end of thrombin generation) and velocity Index (peak height/(time to peak–lag time). LagTime, ttPeak and StartTail are expressed in minutes.

## Data Availability

The original contributions presented in the study are included in the article; further inquiries can be directed to the corresponding authors.

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
