# Peer review of "Assessment of Platelet Aggregation and Thrombin Generation in Patients with Familial Chylomicronemia Syndrome Treated with Volanesorsen: A Cross-Sectional Study"

_biomedicines, 2024, doi:10.3390/biomedicines12092017_

Round 1

Reviewer 1 Report

Comments and Suggestions for Authors

Ilenia Calcaterra et al have presented an effective cross-sectional study. However, due to very less sample size in this study it is hard to derive the outcome of this study.

This work can be accepted after minor corrections.

Specific comments:

1.      Line 19 : correct the statement  (Thormbin)

2.      Line 35 : Please define abbreviations at first place

3.      Write detailed mechanism of action for volanesorsen

4.      Line 77 : correct the reference style

5.      Please correct “ µl” to uL

6.      Please correct “ml” to mL 

Author Response

Reviewer 1

Comment 1: Line 19 : correct the statement  (Thormbin)

Response 1: We want thank reviewer 1 for the suggestion we updated in the manuscript

Comment 2: Line 35 : Please define abbreviations at first place

Response 2: We want thank reviewer 1 for the suggestion we updated in the manuscript

Comment 3: Write detailed mechanism of action for Volanesorsen

Response 3: We want thank reviewer 1 for the suggestion we updated in the manuscript

Comment 4: Line 77: correct the reference style

Response 4: We want thank reviewer 1 for the suggestion we updated in the manuscript

Comment 5: Please correct “ µl” to uL

Response 5: We want thank reviewer 1 for the suggestion we updated in the manuscript

Comment 6: Please correct “ml” to mL 

Response 6: We want thank reviewer 1 for the suggestion we updated in the manuscript

Reviewer 2 Report

Comments and Suggestions for Authors

This study assessed platelet aggregation and thrombin generation in patients with familial chylomicronemia syndrome (FCS) treated with volanesorsen. The results showed no significant changes in the general haemostasis assessed by thrombin generation and in platelet function have been observed in FCS patients receiving volanesorsen. There are still some major concerns.

1. The number of FCS patients is limited and the analysis of platelet aggregation was only conducted in one patient.

2. The analysis of platelet aggregation. Two different concentrations for each agonist except arachidonic acid was used (line 143-145), why the results only presented the data at one concentration (line 215-218)? The concentration unit of each agonist is presented wrongly. For example, “1,8 µM” is “”1.8 µM”.

3. Table 2, The differences in the mean value of Velocity Index, Peak and ETP between patients and control were great, but the P-value is more than 0.05. The reason may be related to the great variance in the control individuals. However, there was not any information about the healthy controls in the study.

4. The first paragraph, line 36-39, the genes related to FCS were listed. why APOC3 was not included? Why the antisense oligonucleotide against APOC3 mRNA volanesorsen was used to treat FCS? The four patients investigated in this study were not related to the mutation of APOC3, but with LPL and APOA5. Whether volanesorsen treatment is effective? No description about the treatment effect was addressed in the text.

5. “,” and “.” were used confusingly. Line 24-26, Line 190, Line 143-145, Line 215-218.

6. Line 132, “CaCl2” was wrong.

7. Line 201, “FTE” should be “EPT”

8. The format of the references was not consistent with the requirement of the journal.

9. Line 72, 77, (Larouche, Brisson et al. 2023) was reference 15. Thus, all references cited in the text must be carefully checked.

10. The discussion was suggested to addressed briefly. The results obtained is relatively less, the discussion is too long and over-interpreted, especially line from 249 to 281.

Author Response

Reviewer 2

This study assessed platelet aggregation and thrombin generation in patients with familial chylomicronemia syndrome (FCS) treated with volanesorsen. The results showed no significant changes in the general haemostasis assessed by thrombin generation and in platelet function have been observed in FCS patients receiving volanesorsen. There are still some major concerns.

Comment 1: The number of FCS patients is limited and the analysis of platelet aggregation was only conducted in one patient.

Response 1: We want to thank reviewer 2 for the comment. Familial Chylomicronemia Syndrome (FCS) is a rare monogenic disease (prevalence of 1-9:1000000) and therefore our sample is representative of disease prevalence in our region according to Italian epidemiological data. Our sample rapresent one of the most large cohort of patients treated with volanesorsen. We agree that another limitation derives from the potential influence of chylomicrons on platelet aggregation test. However, results were entirely comparable to those obtained in healthy controls. It would be interesting to perform the same analyses in patients reporting more severe PLC reduction to explore the presence of functional changes in this setting. We described fully these limitations in discussion.

Comment 2: The analysis of platelet aggregation. Two different concentrations for each agonist except arachidonic acid was used (line 143-145), why the results only presented the data at one concentration (line 215-218)? The concentration unit of each agonist is presented wrongly. For example, “1,8 µM” is “”1.8 µM”.

Response 2: We want thank reviewer 2 for the suggestion we updated in the manuscript.

Comment 3: Table 2, The differences in the mean value of Velocity Index, Peak and ETP between patients and control were great, but the P-value is more than 0.05. The reason may be related to the great variance in the control individuals. However, there was not any information about the healthy controls in the study.

Response 3: We want thank reviewer 2 for the suggestion we updated in the manuscript and we added information about the healthy control group.

Comment 4: The first paragraph, line 36-39, the genes related to FCS were listed. why APOC3 was not included? Why the antisense oligonucleotide against APOC3 mRNA volanesorsen was used to treat FCS? The four patients investigated in this study were not related to the mutation of APOC3, but with LPL and APOA5. Whether volanesorsen treatment is effective? No description about the treatment effect was addressed in the text.

Response 4: We want to thank reviewer 2 for the comment we added a more detailed explanation about APCO-III role in lipid metabolism and Volanesorsen mechanism of action. APO-CIII is not included in causative gene involved in FCS pathogenesis. About effects of Volanesorsen on lipid profile we observed a median % reduction in triglycerides of 60% in our population. I added the median TG values in table 2.

Comment 5: “,” and “.” were used confusingly. Line 24-26, Line 190, Line 143-145, Line 215-218.

Response 5: We want thank reviewer 2 for the suggestion we updated in the manuscript.

Comment 6: Line 132, “CaCl2” was wrong.

Response 6: We want thank reviewer 2 for the suggestion we updated in the manuscript.

Comment 7: Line 201, “FTE” should be “EPT”

Response 7: We want thank reviewer 2 for the suggestion we updated in the manuscript.

Comment 8: The format of the references was not consistent with the requirement of the journal.

Response 8: We want thank reviewer 2 for the suggestion we updated in the manuscript

Comment 9: Line 72, 77, (Larouche, Brisson et al. 2023) was reference 15. Thus, all references cited in the text must be carefully checked.

Response 9: We want thank reviewer 2 for the suggestion we updated in the manuscript.

Comment 10: The discussion was suggested to addressed briefly. The results obtained is relatively less, the discussion is too long and over-interpreted, especially line from 249 to 281.

Response 10: We want thank reviewer 2 for the suggestion we updated in the manuscript

Reviewer 3 Report

Comments and Suggestions for Authors

Content suggestions:

1.         Why did the Authors include thrombin generation test, as this method helps to predominantly the disorders of coagulation ?

2.         Did the patients take any medication interfering with the results ?

Author Response

Comment 1: Why did the Authors include thrombin generation test, as this method helps to predominantly the disorders of coagulation?

Response 1: We want to the reviewer 3 for the comment.  We decided to test thrombin generation to assess global hemostasis. It is also well known that platelets play a major role in localizing and controlling the burst of thrombin generation leading to fibrin clot formation. From the first functional description of platelets, it has been recognized that platelets supply factors that support the activation of prothrombin. In light of this thrombin generation represent a test to full explore platelet function (PMID: 12231555; PMID: 15116259).

Comment 2: Did the patients take any medication interfering with the results?

Response 2: We want to thank reviewer 3 for the question. I confirm that patients did not assume any drug potentially interfering with platelet aggregation and function.

Round 2

Reviewer 2 Report

Comments and Suggestions for Authors

References section. The jounal of some references have the full names and others have abbreviated names.

Author Response

Reviewer 2

Comment 1: References section. The jounal of some references have the full names and others have abbreviated names.

Reaponse 1: We want to thank reviewer 2 for the suggestion, we update bibliography section